# Cycloartane-Type Triterpenes and Botanical Origin of Propolis of Stingless Indonesian Bee *Tetragonula sapiens*

**DOI:** 10.3390/plants8030057

**Published:** 2019-03-08

**Authors:** Niken Pujirahayu, Toshisada Suzuki, Takeshi Katayama

**Affiliations:** 1Laboratory of Biomass Chemistry, Faculty of Agriculture, Kagawa University, Kagawa 761-0795, Japan; rahayuken08@gmail.com (N.P.); t-suzuki@ag.kagawa-u.ac.jp (T.S.); 2Department of Forestry, Faculty of Forestry and Environmental Sciences, Halu Oleo University, Kendari 93232, Indonesia

**Keywords:** *Tetragonula sapiens* bee propolis, Southeast Sulawesi, cycloartane-type triterpene, botanical origin, *Mangifera indica*

## Abstract

This study clarifies the chemical constituents and botanical origin of *Tetragonula sapiens* Cockerell bee propolis collected from Southeast Sulawesi, Indonesia. Propolis samples and resin of *Mangifera indica* were extracted with 99% ethanol to obtain an ethanol extract of propolis (EEP) and an ethanol extract of *M. indica* resin (EEM). Column chromatography, thin-layer chromatography (TLC), and high-performance liquid chromatography (HPLC) were developed and used for the separation and isolation of compounds from the ether-soluble fraction. The structure of the compounds was determined by nuclear magnetic resonance (NMR) spectroscopic analysis, and their molecular weight analyzed by gas chromatography–mass spectrometry (GC–MS). The HPLC chromatogram of the EEP was then compared with the HPLC chromatogram of EEM to investigate the botanical origin of propolis. Five compounds were isolated from the EEP, and their structures were determined as mangiferolic acid, cycloartenol, ambonic acid, mangiferonic acid, and ambolic acid, which are cycloartane-type triterpenes. The characteristic peak of the HPLC chromatograms of EEP and EEM showed a similar pattern, which is that the main components of propolis were also found in *M. indica* resin. These results suggested that the propolis from Southeast Sulawesi was rich in cycloartane-type triterpenes, and the plant source of the propolis could be *Mangifera indica* (mango).

## 1. Introduction

Propolis is one of the natural ingredients produced by honeybees, and it has been widely used as traditional medicine for a long period of time. It is a mixture of beeswax, resin, and other materials that are collected by honeybees like *Tetragonula sapiens* Cockerell, a kind of stingless bee, from plant buds, leaves, and exudates. Bees utilize propolis as building material for their nests as well as tools to prevent the growth of bacteria and fungi in the nest [1,2]. Specifically for stingless bees, propolis is also used to construct storage pots for pollen and honey [3]. 

Propolis contains a variety of compounds, such as (poly)phenols (flavonoids, phenolic acids, and their esters), terpenoids, steroids, amino acids, waxy acid, and sugars [4]. It has been reported that propolis has antioxidant, antimicrobial, antiparasitic [5], antiviral, anti-inflammatory [6,7], and anticancer [8] properties. A recent study also proved that propolis has antitoxic and antimutagenic activities [9]. 

All types of honeybees can produce propolis, but the generated amount of propolis is different depending on the genus or species and the flora of the region. The genus that generates a large amount of propolis is Tetragonula, which belongs to a group of stingless bees of the tribe Meliponini. Unlike Apis bees, Tetragonula bees are smaller than Apis bees, and have a non-functional stinger to defend against nest intruders, but use their jaws to bite them. There are more species of stingless bees than stinging honeybees. There are several hundred species that exist throughout the world that differ significantly in color, body size, and colony size [10,11]. The number of stingless-bee species is estimated to be more than 400 species, much more than that of honeybees that have stingers (11 species) [12]. *T. sapiens* Cockerell is a stingless-bee species in tropical regions. This species is found in Indonesia (Central Sulawesi [13] and Maluku), Philippine, New Guinea, Solomon Island, and Australia (Queensland) [14].

Studies of propolis components from stingless bees are still lacking, even though *Apis mellifera* species have been widely reported, especially in the Southeast Asia region. The propolis of a stingless bee from Brazil, *Tetragonisca angustula,* has triterpenes α-amyrin and lanosterol, and benzophenones are major compounds of *Melipona scutellaris* [15,16]. The propolis of *Tetrigona laeviceps* and *Tetrigona melanoleuca* in Thailand contains a prenylated xanthone, α-mangostin, and dipterocarpol, a dammarane triterpene [17]. The propolis of a Vietnamese stingless bee, *Trigona timor*, contains cycloartane-type triterpenes [18].

Propolis components and properties depend on its plant sources [19]. The cypress is the major plant source of Maltese propolis, and *Populus* bud exudates are the main sources of the propolis of Bologna, Italy, and Algeria. The resin of *Macaranga tanarius* L. and *Mangifera indica* L. is the source of propolis from Japan, Myanmar, and Vietnam [18,20]. *Garcinia mangostana* is the primary source of propolis from Thailand [17]. Trusheva et al. [21] have reported a propolis compound from the *Apis mellifera* honeybee of Indonesia, but the compounds of stingless-bee propolis are unknown, and the primary botanical source of these compounds has not yet been identified, especially in Sulawesi, Indonesia.

This study clarifies the chemical constituents of the Indonesian *T. sapiens* Cockerel bee propolis from Southeast Sulawesi, Indonesia, and their botanical origin.

## 2. Results

### 2.1. Main Compounds of T. sapiens Bee Propolis from Jatibali, South Konawe (P1) and Kendari District (P2), Southeast Sulawesi

The ether-soluble fraction of the *T. sapiens* bee propolis sample was subjected to a series of chromatographic-separation experiments giving two isolated compounds from P1 and three isolated compounds from P2. The former were mangiferolic acid (**1**) [22,23] and cycloartenol (**2**) [23]. The latter were ambonic acid (**3**) [23,24], mangiferonic acid (**4**), and ambolic acid (**5**) [23,25] (Figure 1a,b).

These compounds were identified by comparing their nuclear magnetic resonance (NMR) spectroscopic data (Table 1 and Table 2) and their molecular weight (Table 3) with those in the literature.

Based on the NMR spectra data (Table 1 and Table 2), all isolated compounds were included in the cycloartane triterpenes. A typical cyclopropane moiety characterizes this group. The presence of this bond is seen in the ^13^C spectra and ^1^H spectra (Appendix A).

The results of the mass-spectra calculations of the isolated compounds indicate that these compounds are cycloartane-type triterpenes with key fragmentation and their respective molecular formulas (Table 3).

Compounds **1** and **2** were present in more than 80% of the total area in the chromatogram obtained at 254 nm of the ether-soluble fraction of propolis from South Konawe (Figure 2 and Table 4); these two compounds are the main compounds of propolis P1, while other compounds are seen in small amounts.

The results of the HPLC chromatogram indicate that propolis from Kendari (P2) contains compounds that are more diverse than Jatibali propolis (P1). Compounds **3**, **4**, and **5** dominate propolis from Kendari (P2), with more than 36% of the total area in the chromatogram obtained at 254 nm of the ether-soluble fraction in this propolis (Figure 3 and Table 5).

### 2.2. Botanical Origin of Propolis

The comparison between the main propolis components with the main components of mango resin can be seen in Figure 4A–C.

*M. indica* trees that are around the nest emit reddish-brown sap (Figure 5a,b).

## 3. Discussion

### 3.1. Main Compounds of T. sapiens Bee Propolis from Jatibali, South Konawe (P1) and Kendari District (P2), Southeast Sulawesi

From all isolated compounds, four of them, cycloartenol (**2**), ambonic acid (**3**), mangiferonic acid (**4**), and ambolic acid (**5**), are the first cycloartane-type triterpenes found in Indonesian propolis (both Apidae and Melliponi bee propolis). A previous study conducted by Trusheva et al. [21] reported that other cycloarten-type triterpenes, mangiferolic acid, isomangiferolic acid, and 27-hydroxyimangiferolic acid had been found in propolis from East Jawa, Indonesia, different regions than Sulawesi, and different bees (*Apis mellifera*), but these were not Compounds **2**, **3**, **4**, or **5**.

The constituent of this propolis is very similar to the components of Cameroonian (*A. mellifera*) propolis. All cycloartane-type triterpenes in this propolis were also found in the Cameroonian propolis. Some propolis from other tropical regions were also reported to contain cycloartane-type triterpenes. Myanmar (*A. mellifera*) propolis and Vietnam Trigona propolis are rich in cycloartane-triterpene compounds, both of which contain mangiferonic acid, 27-hydroxymangiferonic acid, and mangiferolic acid as the main constituents [26]. Malaysian propolis (*Trigona itama*) has cycloartenol [27]. Brazilian propolis contains cycloartane-type triterpenes, mangiferolic acid, 3β-hydroxy-24-methylenecycloartane-26-oic acid, and ambonic and ambolic acid [25]. Thailand propolis contains dammarane triterpene, dipterocarpol, ursolic acid, and ocotillone [17].

It is interesting that, although the botanical origin of propolis from Myanmar, Vietnam, Malaysia, and Brazil is the same plant, *M. indica*, the cycloartane-type triterpenes contained in that propolis are not the same. Ambonic and ambolic acid are only found in Cameroonian propolis, Teresina (Brazil), and this propolis. This difference may be due to the *M. indica* variety is not the same from each region where the propolis originates.

P1 and P2 propolis showed different major compounds. Mangiferolic acid and cycloartenol acids are the major compounds in propolis from South Konawe; more than 80% of the ether fraction of this propolis is dominated by these two compounds (Figure 2 and Table 4). Ambonic, mangiferonic, and ambolic acid, meanwhile, are three compounds that are mostly present in propolis from the Kendari district (Figure 3 and Table 5). This difference may be due to the behavior of bees in collecting resins in colonies, species and varieties of plants, as well as differences in the parts and amount of resin taken. The composition of the secondary metabolites contained in each part of the plant such as shoots, leaves, branches, and stems is not the same even in one plant/tree. Bees can collect resin from the bud exudates or the sap that comes out of the wound on a branch or tree trunk. These results also indicate that the differences found between propolis samples from various regions are mainly due to differences in the flora and less to the species of the bees.

The study of bee propolis and its plant origin revealed a unique relationship between insects and plants. A study reported by Leonhardt [28,29] on six stingless-bee species (*Tetragonilla collina*, *Tetragonula melanocephala*, *T. geissleri/laeviceps*, *T. melanocephala*, *Lepidotrigona terminata*, *Pariotrigona pendleburyi*) in Borneo, Malaysia, found 113 terpenes in the nest, and 83 of them were also found on the surface of some of the bees’ bodies. The terpenes were found to consist of sesquiterpenes and triterpenes, which represent the most prominent terpene class. This is a characteristic of the dipterocarp tree, the dominant family tree of Southeast Asian forests. The *T. melanocephala*, L. *terminata*, *P. pendleburyi*, and *T. fuscobalteata* species do not even have sesquiterpenes at all, but all species have triterpenes.

This study revealed the same thing about main propolis components from *T. sapiens* nests from the Sulawesi region, namely, triterpene. Even more uniquely, the main components of this propolis are in the same group of compounds, cycloartane-type triterpenes. Therefore, it is possible that stingless-bee species can specifically ‘filter’ resin derivative compounds, with some species not included in all compound classes, suggesting that terpene acquisition has a genetic basis in these bees [29]; specifically for this *T. sapiens,* it is triterpene.

### 3.2. Botanical Origin of T. sapiens Propolis

Compounds **1** and **2** were present in considerably more than 80% of the total area in the chromatogram obtained at 254 nm of the ether-soluble fraction of the propolis from South Konawe (Table 4), and Compounds **3**, **4**, and **5** were more than 36% present of the total area in the chromatogram obtained at 254 nm of the ether-soluble fraction of the propolis from Kendari (Table 5). It is known that all cycloartane-type triterpenes in this propolis are also present in *M. indica* resin as its most common components [22]. This result was also confirmed by the peak HPLC chromatogram characteristic of the EEP and EEM, which showed the same pattern (Figure 4). It was concluded that the main propolis component came from *M. indica* resin. These results suggest that the predominant plant source of propolis in both Jatibali, South Konawe, and Kendari, Southeast Sulawesi could be *M. indica* (mango, Figure 5), which is widely present around where these bees’ hives are located. By our observations, the *T. sapiens* bees frequently visited these plants.

This study characterized the botanical origin of propolis from Sulawesi for the first time. Although some of the components in this propolis have not been identified, the major compounds isolated from the propolis sample indicate that the main plant source is *M. indica.* Interestingly, some studies also show that some of the main propolis components are also present in mango resins. That is especially true for both Apis bee propolis and stingless-bee (Meliponi) propolis from the tropics. The botanical origin of *A. mellifera* bee propolis from Myanmar [26], Cameroon [23], and Indonesia [21] is reported to be *M. indica.* Likewise, the propolis of stingless bee *T. spinipes* from Brazil [30], *Trigona itama* from Malaysia [27], *T. minor* from Vietnam [18], and *Tetragonula biroi* from the Philippines, [31] was reported to have *M. indica* as its primary source.

In the tropics, many plants produce resins, and bees can collect these resins from various species around the nest, but several studies show that bees exhibit selective behavior and prioritize certain plant species during resin collection [32,33]. In this study, besides *M. indica*, several plants around the nest, such as *Anacardium occidentale*, *Artocarpus cempedan*, *Euphorbia milii*, *Euphorbia pulcherima*, and some flowering plants were also seen as visited by *T. sapiens* bees. However, the primary source of this propolis is the resin and bud exudates of *M. indica*, and it is convincing that *M. indica* is an important plant source of propolis in the tropics, not only in Asia but also in Africa and Central America.

Several studies have shown that the botanical origin of stingless-bee propolis in Southeast Asia forests is Dipterocarpaceae, which are rich in terpenes [17,28]. In this study, however, it was demonstrated that the origin of plant propolis is mango (Anacardiaceae). It seems that bees instinctively/genetically have the ability to recognize types of compounds in resin plants (in this case, triterpenes). So, what bees are looking for is not only focused on plant species, but also on specific compounds that exist in plant resins. The plant origin of propolis can be different, but its main components have similarities or are in the same group. Propolis samples in this study were obtained from apiaries around settlements and plantations where there are many mango plants. Thus, bees take advantage of the existence of this plant to collect resins and triterpenes to build their nests. Triterpenes have been known as secondary metabolites that have extensive biological activity. These compounds are mainly synthesized by plants to protect themselves from various attacks by disrupting organisms and diseases. Then, bees collect, choose, and use them to form propolis as a protective material for nests and colony members, a fascinating bee–plant interaction.

There have been some studies on the biological activity of these cycloartane-type triterpenes (23-hydroxyisomangiferolic acid) from Vietnam stingless-bee propolis (*Trigona timor*) and isoambolic acid from *M. indica,* which has strong preferential cytotoxicity to human pancreatic PAN-1 cancer cells [18,34]. Further research is needed to examine other biological activities of the isolated compounds in this study so more information could be obtained from this propolis and use it in the field of drug development from natural products.

## 4. Materials and Methods

### 4.1. Materials

Propolis samples were collected from the beehives of *Tetragonula sapiens* Cockerell, and *Mangifera indica* resin was collected from *M. indica* trees from September to November 2016 in Jatibali, South Konawe district (P1) and Kendari district (P2), Southeast Sulawesi, Indonesia (Figure 6a,b). They were identified by the Research Center for Biology, Indonesian Institute of Sciences, Bogor, Indonesia based on bee specimens, nest structures (Figure 7a) and the entrance tube form (Figure 7b). The two voucher samples of raw propolis, JK 0231 (South Konawe propolis) and JK 0232 (Kendari propolis), and the resin of *M. indica* (JK 0121) were deposited at the Faculty of Forestry and Environmental Sciences, Halu Oleo University, Kendari, Southeast Sulawesi, Indonesia.

### 4.2. Propolis and M. Indica Resin Extraction

#### 4.2.1. Propolis from Jatibali, South Konawe District (P1)

The raw P1 propolis sample (10 g) was ground to a fine powder that was extracted three times with 50 mL of 99% ethanol in a shaker at room temperature for 24 h. After filtration of each extract with a filter paper (Advantec no. 2, 150 mm), filtrates were combined, and this solution (EEP) was evaporated in vacuo with a rotary evaporator (Eyela CCA-1111, Tokyo Rikakikai, Tokyo, Japan). The P1 EEP (5.66 g) was partitioned between diethyl ether (50 mL) and ultrapure water (50 mL) with a separatory funnel to give an ether-soluble fraction (4.87 g) and an aqueous layer. The latter was again extracted with EtOAc, and the EtOAc-soluble fraction (0.09 g) and the aqueous fraction (0.56 g) were obtained. The ether-soluble and ethyl acetate-soluble fractions were washed with a saturated NaCl solution; then, the organic layer was dried over hydrous Na_2_SO_4_ before concentration.

#### 4.2.2. Propolis from Kendari District (P2)

The raw P2 propolis sample (15 g) was ground to a fine powder that was extracted with the same method as P1 to obtain the P2 EEP. The EEP was partitioned between diethyl ether (50 mL) and ultrapure water (50 mL) with a separatory funnel to give an ether-soluble fraction (7.35 g) and an aqueous layer. The latter was again extracted with EtOAc to obtain an EtOAc-soluble fraction (0.51 g) and aqueous fraction (0.57 g).

#### 4.2.3. *M. Indica* Resin

The *M. indica* resin sample (2.0 g) was ground into a fine powder then extracted three times with 20 mL 99% ethanol in a shaker at room temperature for 24 h. After filtration using the same method as P1, an ethanol extract from *M. indica* resin (EEM, 1.176 g) was obtained. The EEM was then applied to take HPLC (UV 254 nm) profiles and compared with P1 and P2 EEP.

### 4.3. TLC and HPLC Analysis

Analytical thin-layer chromatography (TLC) and preparative TLC were performed on silica gel 60 F_254_ glass plates (20 × 20 cm, thickness of 0.25 and 0.5 mm, respectively (Merck, Damstardt, Germany)). Detection of the spot and bands was achieved under a UV-light shortwave of 254 nm (UVP, UVG-54, Upland, CA, USA). The bands corresponding to compounds were scratched off and eluted with a mixture of dichloromethane and MeOH (8:2). Analytical HPLC was performed with an HPLC system (a Jasco PU-980 intelligent HPLC Pump, a Jasco MD-2010 plus multiwavelength detector, a Jasco UV-970 intelligent UV/VIS detector, and a Jasco CO-965 column oven with a COSMOSIL packed column 5C-18-MS-II (ID 4.6 × 250 mm). Ten microliters of a solution of the propolis extract (1000 ppm) in MeOH, filtered through a 0.45 μm membrane filter, was injected into the injector of the HPLC system. The eluents were (A) 60% MeOH and (B) 100% MeOH. The eluents were filtered through a 0.5 μm membrane filter (PTFE). Separation was performed at 40 °C at a flow rate of 1 mL/min for 50 min (retention time). Preparative HPLC was done using the same system as above with a different column size (COSMOSIL packed column 5C18-AR-II (ID 10 × 250 mm; pressure 54 kg/cm^2^). Fractional separation was performed at 40 °C at a flow rate of 2 mL/min with the following gradient: from (A) 80% MeOH to (B) 100% MeOH in 90 min. Column chromatography was performed with an FMI pump system (Yamazen, Osaka, Japan) with a column of silica gel 60 (Merck, 0.04–0.063 mm, 230–400 mesh), using a gradient system with ethyl acetate and *n*-hexane.

### 4.4. NMR and Gas Chromatography–Mass Spectrometry (GC–MS) Analysis

NMR spectra (^1^H, ^13^C, ^1^H COSY, HMBC, and HMQC) were recorded on a JEOL JNM ECA 600 FT-NMR spectrometer (600 MHz for ^1^H and 150 MHz for ^13^C) using chloroform-d (99.8% D) as a solvent, with 0.03% tetramethylsilane (TMS) as an internal standard. Chemical shifts were expressed as δ (in ppm), and coupling constants (*J*) were recorded in Hertz.

The molecular weight of the compounds was analyzed by GC–MS after silylation of the samples, which was done with a QP2010 SE gas chromatograph mass spectrometer (Shimadzu, Tokyo, Japan). The device was equipped with an Intercap 5MS fused silica capillary column (30 mm × 0.25 mm I.D. and 0.25 µm film thickness), with an electronic pressure control module and a split/splitless injector (AOC 20i, Shimadzu). GC temperature was programmed from 100 to 310 °C at a rate of 5 °C/min, with helium as a carrier gas. The pressure was 100 kPa, split ratio was 1:80, the flow rate through the column was 1.33 mL/min in constant flow mode, and the ionization voltage was 70 eV. Approximately 1 mg of the propolis sample was mixed with 50 µL of dry pyridine and 75 µL of bis(trimethylsilyl) trifluoroacetamide. The mixture was heated at 80 °C for 20 min, and 1 µL of the sample was injected.

### 4.5. Isolation Compounds of Propolis

The ether-soluble fraction of P1 propolis was subjected to column chromatography on silica gel 60 (Merck, 40–60 μm, 230–400 mesh) with stepwise elution of EtOAc/*n*-hexane (50%, 70%, 90%, and 100%). The eluates were concentrated in vacuo with a rotary evaporator to afford 5 major fractions (Fraction 1, 0.26 g; Fraction 2, 2.10 g; Fraction 3, 0.98 g; Fraction 4, 1.12 g; Fraction 5, 1.20 g). Fraction 3 (977 mg) was dissolved in a small amount of EtOAc and subjected to TLC (EtOAc/*n*-hexane = 1:1 (*v*/*v*)) to give three subfractions (Fraction 3.1, 0.12 g; Fraction 3.2, 0.83 g; Fraction 3.3, 0.09 g). Subfraction 3.2 (830 mg) was purified by preparative HPLC (eluent: MeOH/H_2_O = 80:20 (*v*/*v*)), followed by repeated TLC and HPLC separation, to yield 4 sub-subfractions, Fraction 3.2.1, Fraction 3.2.2, Fraction 3.2.3, and Fraction 3.2.4. Fraction 3.2.4 (50.1 mg) was recrystallized with MeOH/acetone (1:1, *v*/*v*) to give **1** (20.6 mg).

Fraction 2 (2.1 g) from the ether extract of P1 was purified by preparative TLC with EtOAc/*n*-hexane (30:70, *v*/*v*) to give 5 subfractions (2.1–2.5), in which Subfraction 2.3 (484.1 mg) was separated by preparative TLC with EtOAc/*n*-hexane (30:70, *v*/*v*) to afford 3 subfractions (2.3.1–2.3.3), in which Fraction 2.3.1 (118 mg) was recrystallized with MeOH/acetone (1:1, *v*/*v*) to give **2** (5.9 mg). The scheme of successive fractionations of *T. sapiens* of P1 is shown in Appendix A.

The ether soluble fraction of P2 propolis was subjected to column chromatography using the same method as P1. P2 eluates were concentrated in vacuo with a rotary evaporator to afford 5 major fractions (Fraction 1, 29 g; Fraction 2, 1.37 g; Fraction 3, 1.09 g; Fraction 4, 0.95 g; Fraction 5, 0.56 g). Fraction 3 (1.09 g) was dissolved in a small amount of EtOAc and subjected to TLC (EtOAc/*n*-hexane = 4:6 (*v*/*v*)) to afford four subfractions (Fraction 3.1, 10.9 mg; Fraction 3.2, 254.4 mg; Fraction 3.3, 793.8 mg, and Fraction 3.4, 114.0 mg). Subfraction 3.2 (254.4 mg) was recrystallized with MeOH/acetone (1:1, *v*/*v*) to give an inseparable mixture (137 mg) of **3** and **4** (almost 1:1 ratio).

Fraction 4 (0.95 g) from P2 was purified by preparative RP-TLC with 100% MeOH, to give 4 subfractions (4.1–4.4), in which Subfraction 4.3 (600.1 mg) was separated by preparative TLC with EtOAc/*n*-hexane (30:70, *v*/*v*) to give three subfractions (4.31–4.3.3), in which Fraction 4.3.1 (118 mg) was recrystallized with MeOH/acetone (1:1) to give an inseparable mixture (24.8 mg) of **1** and **5** (almost 2:1 ratio). The scheme of successive fractionations of *T. sapiens* of P2 is shown in Appendix A.

Compound **1** (mangiferolic acid): white amorphous solid; Rf 0.40 in EtOAc/*n*-hexane (4:6); MS-EI: *m*/*z* 600 [M+2TMS]^+^, (calculated molecular weight for C_30_H_48_O_3_, 456.71); ^1^H NMR (CDCl_3_) and ^13^C NMR (CDCl_3_) (see Table 1 and Table 2).

Compound **2** (cyloartenol): white amorphous solid; Rf 0.45 in EtOAc/*n*-hexane (3:7); MS-EI: *m*/*z* 570 [M+2TMS]^+^, (calculated molecular weight for C_30_H_50_O, 426.72) ^1^H NMR (CDCl_3_) and ^13^C NMR (CDCl_3_) (see Table 1 and Table 2).

Compound **3** (ambonic acid): white amorphous solid; Rf 0.5 in in EtOAc/*n*-hexane (4:6); MS-EI: *m*/*z* 612 [M+2TMS]^+^, (calculated molecular weight for C_30_H_50_O, 469.36) ^1^H NMR (CDCl_3_) and ^13^C NMR (CDCl_3_) (see Table 1 and Table 2).

Compound **4** (mangiferonic acid): white amorphous solid; Rf 0.5 in in EtOAc/*n*-hexane (4:6); MS-EI: *m*/*z* 598 [M+2TMS]^+^, (calculated molecular weight for C_30_H_50_O, 454.37) ^1^H NMR (CDCl_3_) and ^13^C NMR (CDCl_3_) (see Table 1 and Table 2).

Compound **5** (ambolic acid): white amorphous solid; Rf 0.35 in 100% MeOH (RP-TLC); MS-EI: *m*/*z* 616 [M+2TMS]^+^, (calculated molecular weight for C_30_H_50_O, 472.38) ^1^H NMR (CDCl_3_) and ^13^C NMR (CDCl_3_) (see Table 1 and Table 2).

## 5. Conclusions

The major compounds of *T. sapiens* bee propolis from Southeast Sulawesi, Indonesia were isolated and identified as five cycloartane-type triterpenes, namely, mangiferolic acid, cycloartenol, and ambonic, mangiferonic, and ambolic acid. The plant source of the propolis from Southeast Sulawesi was suggested to be *M. indica*.

## Figures and Tables

**Figure 1 plants-08-00057-f001:**
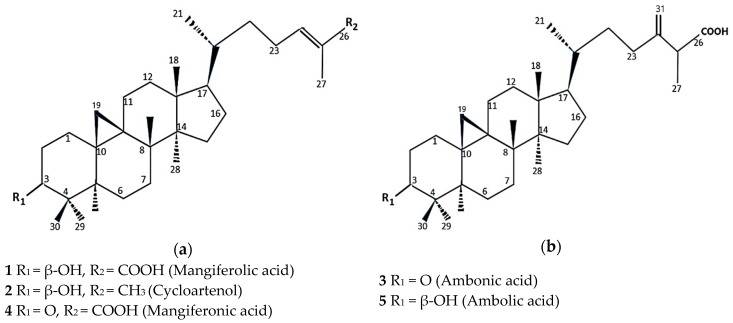
Structures of isolated compounds from *T. sapiens* bee propolis in Southeast Sulawesi (**a**) compounds 1, 2, and 3; (**b**). Compounds 3 and 5.

**Figure 2 plants-08-00057-f002:**
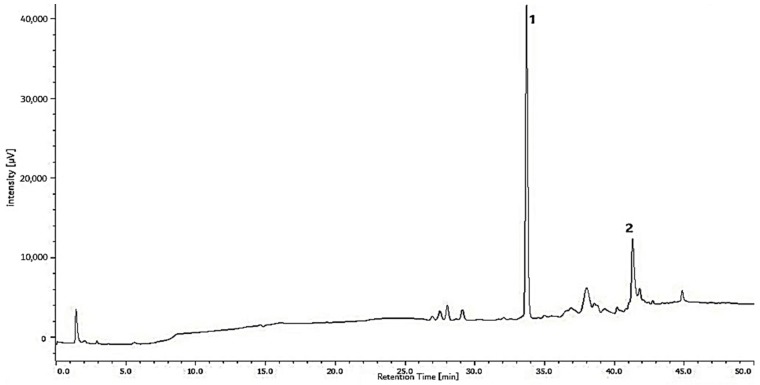
High-performance liquid chromatography (HPLC) chromatogram of the ether fraction of P1 (*T. sapiens* propolis from Jatibali, South Konawe district, Southeast Sulawesi); (**1**) mangiferolic acid and (**2**) cycloartenol.

**Figure 3 plants-08-00057-f003:**
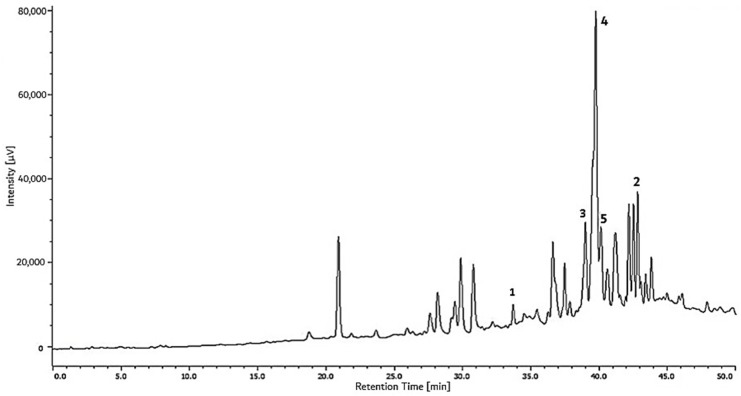
HPLC chromatogram of ether fraction of P2 (*T. Sapiens* propolis from Kendari district, Southeast Sulawesi); mangiferolic acid (**1**), cycloartenol (**2**), ambonic acid (**3**), mangiferonic acid (**4**), and ambolic acid (**5**).

**Figure 4 plants-08-00057-f004:**
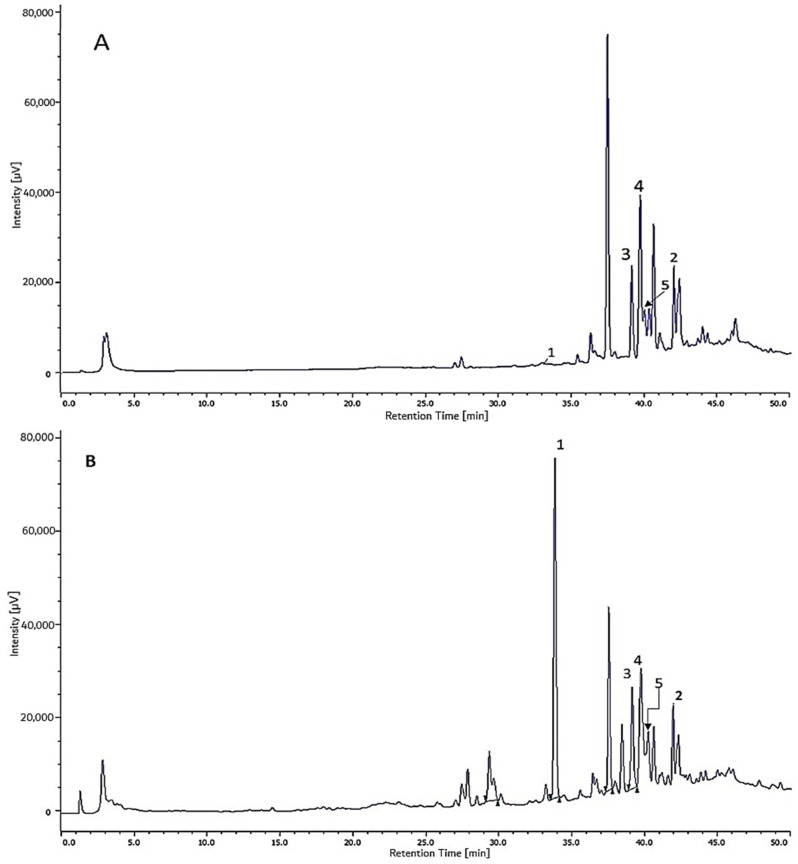
HPLC chromatogram: (**A**) ethanol extract of *M. indica* (EEM), (**B**) Ethanol extract of propolis (EEP) of P1 (*T. sapiens* Konawe Propolis), (**C**) EEP of P2 (*T. Sapiens* Kendari propolis).

**Figure 5 plants-08-00057-f005:**
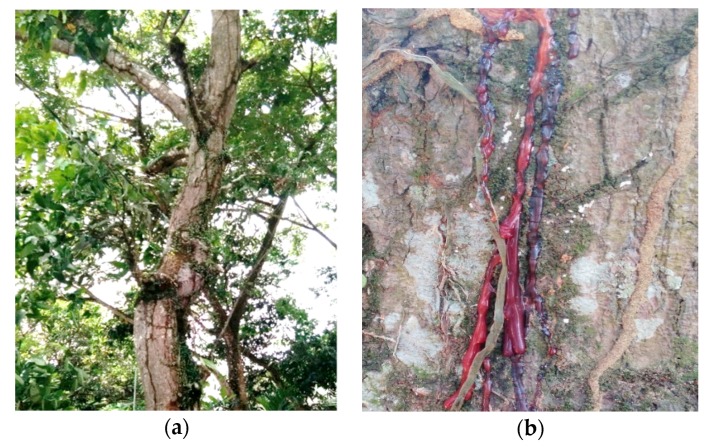
(**a**) *Mangifera indica* tree around *T. sapiens* bee hives in South Konawe and Kendari, Southeast Sulawesi, and (**b**) *M. indica* resin.

**Figure 6 plants-08-00057-f006:**
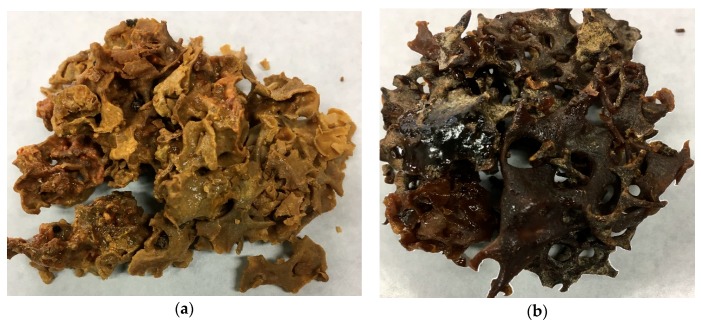
*T. sapiens* propolis from (**a**) South Konawe district and (**b**) Kendari district.

**Figure 7 plants-08-00057-f007:**
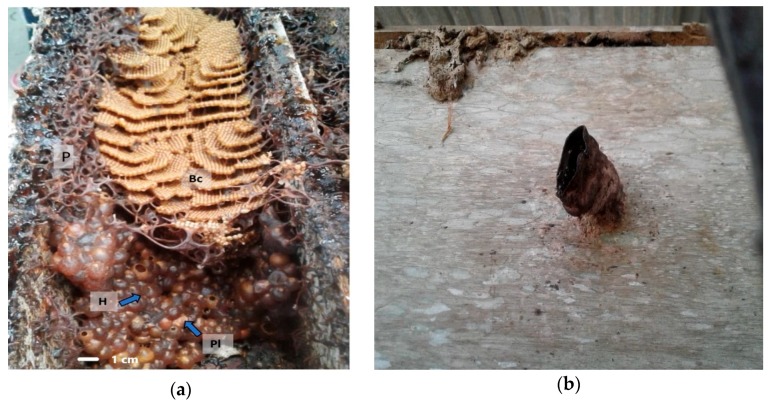
(**a**) *T. sapiens* nest: honey pots (H), pollen pot (Pl), brood cells (Bc), and propolis (P); (**b**) entrance tube of *T. sapiens* nest.

**Table 1 plants-08-00057-t001:** ^13^C nuclear magnetic resonance (NMR, 150 MHz) data in CDCl_3_ (δ in ppm) for isolated compounds from *T. sapiens* bee propolis in Southeast Sulawesi.

Carbon	Mangiferolic Acid (1)	Cycloartenol (2)	Ambonic Acid (3)	Mangiferonic Acid (4)	Ambolic Acid (5)
1	31.96	31.96	33.43	33.43	31.97
2	30.36	30.39	37.48	37.48	30.36
3	78.86	78.86	216.74	216.74	78.88
4	40.48	40.49	50.25	50.25	40.48
5	47.09	47.09	48.43	48.43	47.09
6	21.11	21.11	21.51	21.51	21.11
7	26.43	26.48	25.86	25.86	26.45
8	47.96	48.49	47.87	47.87	47.98
9	19.96	20.01	21.06	21.06	19.97
10	26.07	26.02	25.97	25.97	26.01
11	26.44	26.48	26.69	26.69	26.45
12	32.89	32.89	32.79	32.79	32.91
13	45.34	45.28	45.38	45.38	45.35
14	48.81	48.80	48.75	48.75	48.82
15	35.54	35.58	35.54	35.54	35.55
16	28.15	28.14	28.15	28.15	28.15
17	52.20	52.28	52.21	52.21	52.21
18	18.07	18.03	18.11	18.11	18.07
19	29.89	29.90	29.57	29.57	29.90
20	35.97	35.88	45.37	45.37	35.98
21	18.10	18.23	18.28	18.28	18.11
22	34.79	35.01	34.7	34.7	34.8
23	26.00	24.94	31.63	25.8	25.91
24	145.74	125.26	148.68	145.69	148.67
25	126.42	130.91	45.37	126.63	45.37
26	172.01	17.65	179.80	172.82	179.78
27	12.02	25.74	16.32	11.99	16.32
28	19.31	19.31	19.29	19.29	19.32
29	25.44	25.44	22.17	22.17	25.44
30	14.00	14.00	20.78	20.78	14.02
31			111.03		111.08

**Table 2 plants-08-00057-t002:** ^1^H NMR (600 MHz) data in CDCl_3_ (δ in ppm) for isolated compounds from *T. sapiens* bee propolis in Southeast Sulawesi.

Proton	Mangiferolic Acid (1)	Cycloartenol (2)	Ambonic Acid (3)	Mangiferonic Acid (4)	Ambolic Acid (5)
1	1.62 m, 1.24 m	1.62 m, 1.24 m	1.85 m, 1.54 m	1.85 m, 1.54 m	1.62 m, 1.24 m
2	1.75 m, 1.52 m	1.75 m, 1.52 m	2.73 m, 2.32 m	2.73 m, 2.32 m	1.75 m, 1.52 m
3	3.29 m	3.29 m	-	-	3.29 m
4	-	-	-	-	-
5	1.33 m	1.33 m	1.71 (d, 7.1)	1.71 (d, 7.1)	1.33 m
6	1.49 m, 0.78 m	1.49 m, 0.78 m	1.57 m, 0.94 m	1.57 m, 0.94 m	1.49 m, 0.78 m
7	1.31 m, 1.12 m	1.31 m, 1.12 m	1.38 m, 1.14 m	1.38 m, 1.14 m	1.31 m, 1.12 m
8	1.55–1.62 m	1.55–1.62 m	1.56–1.59 m	1.56–1.59 m	1.55–1.62 m
9	-	-	-	-	-
10	-	-	-	-	-
11	2.03 m, 1.16 m	2.03 m, 1.16 m	1.97 m, 1.15 m	1.97 m, 1.15 m	2.03 m, 1.16 m
12	1.61–1.62 m, 2H	1.61–1.62 m, 2H	1.15–1.21 m, 2H	1.15–1.21 m, 2H	1.61–1.62 m, 2H
13	-	-	-	-	-
14	-	-	-	-	-
15	1.28–1.32 m, 2H	1.28–1.32 m, 2H	1.32–1.34 m, 2H	1.32–1.34 m, 2H	1.28–1.32 m, 2H
16	1.90 m, 1.27 m	1.90 m, 1.27 m	1.92–1.97 m	1.92–1.97 m	1.90 m, 1.27 m
17	1.58–1.61 m	1.58–1.61 m	1.68–1.72 m	1.68–1.72 m	1.58–1.61 m
18	0.97 s	0.97 s	1.00 s	1.00 s	0.97 s
19	0.32 (d, 4.1), 0.54 (d, 4.1)	0.32 (d, 4.1), 0.54 (d, 4.1)	0.79 (d, 3.6), 0.58 (d, 3.4)	0.79 (d, 3.6), 0.58 (d, 3.4)	0.32 (d, 3.5), 0.54 (d,3.5)
20	1.28–1.32 m	1.28–1.32 m	1.41–1.44 m	1.41–1.44 m	1.28–1.32 m
21	0.91 (d 6.6, 3H)	0.91 (d 6.6, 3H)	(d, 6.6 3H)	(d, 6.6, 3H)	(d, 6.6, 3H)
22	1.55 m, 1.16 m	1.55 m, 1.16 m	1.58 m, 1.16 m	1.58 m, 1.16 m	1.55 m, 1.16 m
23	2.26 m, 2.16 m	2.26 m, 2.16 m	2.02 m, 1.92 m	2.02 m, 1.92 m	2.12 m, 1.95 m
24	6.88 (t, 7.3)	5.10 (t, 7.1)	-	6.90 (bt,7.4)	-
25	-	-	3.16 (bq, 6.8)	-	3.15 (bq, 6.8)
26	-	1.55 s	-	-	-
27	1.84 s	1.05 s	1.84 s	1.84 s	1.84 s
28	0.89 s	0.89 s	0.91 s	0.91 s	0.89 s
29	0.97 s	0.97 s	1.05 s	1.05 s	0.97 s
30	0.81 s	0.82 s	1.10 s	1.10 s	0.81 s
31			4.97 bs, 4.93 bs		4.97 bs, 4.93 bs

**Table 3 plants-08-00057-t003:** Molecular ion and key fragmentation on mass spectra of identified compounds in *T. sapiens* bee propolis from Southeast Sulawesi.

Compound Name	Retention Time (min)	Molecular Formula	Key Fragmentation (*m*/*z*)
Cycloartenol (**2**)	41.99	C_30_H_50_O	*m*/*z* 570 (M+2TMS)^+^, 483 (M+2TMS-CH_3_)^+^, 408, 393, 135, 109, 95, 73, 69
Ambonic acid (**3**)	47.64	C_31_H_49_O_3_	*m*/*z* 612 (M+2TMS)^+^, 540 (M+2TMS-CH_3_)^+^, 394, 313, 189, 175, 107, 95, 73
Mangiferonic acid (**4**)	48.67	C_30_H_47_O_3_	598 (M+2TMS)^+^, 526 (M+2TMS-CH_3_)^+^, 421, 388, 311, 189, 133, 107, 95, 73
Mangiferolic acid (**1**)	49.13	C_30_H_49_O_3_	*m*/*z* 600 (M+2TMS)^+^, 585 (M+2TMS-CH_3_)^+^, 510, 495, 467,135, 95, 73, 44
Ambolic acid (**5**)	49.14	C_31_H_49_O_3_	*m*/*z* 616 (M+2TMS)^+^, 585 (M+2TMS-CH_3_)^+^, 510, 495, 467, 135, 95, 73

**Table 4 plants-08-00057-t004:** Distribution of cycloartane-type triterpenes in the *T. sapiens* bee P1 propolis (from Jatibali, South Konawe district, Southeast Sulawesi).

Compound Number	Compound Name	Retention Time (min)	Wave Length (nm)	Area (%)
**1**	Mangiferolic acid	33.68	254	80.25
**2**	Cycloartenol	42.15	254	8.87

**Table 5 plants-08-00057-t005:** Distribution of cycloartane-type triterpenese in the *T. sapiens* bee P2 propolis (from Kendari district, Southeast Sulawesi).

Compound Number	Compound Name	Retention Time (min)	Wave Length (nm)	Area (%)
**3**	Ambonic acid	38.97	254	11.32
**4**	Mangiferonic acid	39.72	254	22.6
**5**	Ambolic acid	40.09	254	2.9

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
