# Peer review of "Cycloartane-Type Triterpenes and Botanical Origin of Propolis of Stingless Indonesian Bee Tetragonula sapiens"

_plants, 2019, doi:10.3390/plants8030057_

Round 1

Reviewer 1 Report

The article entitled “Cycloartane-Type Triterpenes and Propolis Botanical Origin of Stingless Indonesian Bee Tetragonula sapiens” reports the chemical characterization of main components and botanical origin of propolis from a specific stingless Indonesian bee. The authors collected propolis samples, which were extracted with ethanol, and isolated the main components of an ether-soluble fraction through several chromatographic-separation techniques. Then, they compared the HPLC chromatogram of this fraction with the one of ethanolic extract of Mangifera indica and found out that the propolis from Southeast Sulawesi was rich in cycloartane-type triterpenes, and the plant source of the propolis could be M. indica (mango).

In my opinion, the article is well presented, and the used techniques were well employed. Although five triterpenes were reported as main components of this specific propolis for the first time, their structures have been already reported in the literature. So, the novelty of this work is only moderate. However, it should be published on Plants after minor revisions:

- line 19, “of the compounds” should be deleted

- lines 22 and 23, the compounds should not be numbered in the abstract

- line 23, “which are” instead of “which were”

- line 38, “(poly)phenols” instead of “(poly) phenols”

- line 46, the authors used the word “sting” (UK) in singular form, however, they used “stringers” (US) for the plural form (line 50), so, it should be “stinger” (US) instead of “sting” in the singular form for the sack of coherence

- line 55, “α-amyrin” instead of “α-amyrine”

- line 57, “dipterocarpol, a dammarane triterpene” instead of “dipterocarpol dammarane triterpene”

- line 73, “The ether-soluble fraction” instead of “ether extract”

- In Figure 1, the authors subtitled the structures with the letters (a) and (b), but they did not mention them in the text. In the same figure, mangiferonic acid should not be in italics

- In Table 1, the heading “Position” should be replaced by “Carbon”

- line 82, “Based on the NMR spectra data (Tables 1 and 2)” instead of “Based on the NMR spectrum of the 1H and 13C tables”

- line 83, “cyclopropane moiety” instead of “C9-C19-C10 bond”

- line 84, “ring can be seen in the 1H and 13C NMR spectra” instead of “bond is seen in the 1H spectra and 13C spectra”

- lines 84 and 85, the authors refer to carbon 19, however, they present the chemical shifts of H-19 protons. They should rephrase this statement and present this important data in a better way.

- In Table 2, the heading “Position” should be replaced by “Proton”

- In the same table, when the authors assign a proton as a multiplet, they present one or two chemical shift values. However, in 1H NMR spectra, a multiplet should correspond to a range of chemical shifts (e.g., 1.72-2.01 ppm).

- In the same table, why did the authors present two chemical shift values for dd of H-19 protons and for bs of H-31 protons? If the protons of these two methylene groups are not equivalents and there are two signals corresponding to one carbon, this should be referred in the table.

- In the same table, why did not the authors present the coupling constants corresponding to d, dd, t and so on?

- Were compounds 3 and 4 obtained as a mixture? In the SI, the authors present just one 1H and 13C spectrum for both compounds. If they were isolated as a mixture, this should be referred somewhere in the text.

- lines 130 and 132, did the authors mean “cycloartane-type” instead of “cycloarten-type”?

Author Response

Point 1: line 19, “of the compounds” should be deleted

Respone 1: yes, it was deleted

Point 2: lines 22 and 23, the compounds should not be numbered in the abstract

Respone 2: yes it was deleted, although in some journal numbers compounds are written

Point 3: line 23, “which are” instead of “which were”

Respone 3: yes it has been changed

Point 4: line 38, “(poly)phenols” instead of “(poly) phenols”,

Respone 4: ok

Point 5: line 46, the authors used the word “sting” (UK) in singular form, however,

they used “stringers” (US) for the plural form (line 50), so, it should be

“stinger” (US) instead of “sting” in the singular form for the sack of coherence

Point 14: lines 84 and 85, the authors refer to carbon 19, however, they present the

chemical shifts of H-19 protons. They should rephrase this statement and

present this important data in a better way.

Respone 14: yes, this is mistake, we will rephrase the statement

Point 15: In Table 2, the heading “Position” should be replaced by “Proton”,

Respone 15: ok, it has been changed

Point 16: In the same table, when the authors assign a proton as a multiplet, they

present one or two chemical shift values. However, in 1H NMR spectra, a

multiplet should correspond to a range of chemical shifts (e.g., 1.72-2.01 ppm).

Respone 16: We have carefully assigned the signal from the spectral data and obtained that value so that the writing can be like that. And we also compared the spectra of 1H NMR from several papers for the same compound that they wrote protons as multiplets, so we argued that one chemical shift value could be written, but some of them we write with a range as you suggested.

Point 17: In the same table, why did the authors present two chemical shift values for

dd of H-19 protons and for bs of H-31 protons? If the protons of these two methylene groups are not equivalents and there are two signals corresponding to one carbon, this should be referred in the table.

Respone 17: Thank you, we have completed the chemical shift value for the symbol d, (some mistake it is not dd but xx d, yy d)

Point 18: In the same table, why did not the authors present the coupling constants

corresponding to d, dd, t and so on?

Respone 18: we have completed the chemical shift value for the symbol d, t, bq

Point 19: Were compounds 3 and 4 obtained as a mixture? In the SI, the authors

present just one 1H and 13C spectrum for both compounds. If they were isolated as a mixture, this should be referred somewhere in the text.

Respone 19: It was referred in method (line 319)

Point 20: lines 130 and 132, did the authors mean “cycloartane-type” instead of

“cycloarten-type”?

Respone 20: yes it has been changed

Thank you very much for your correction and suggestions for improving this manuscript

Regards,

Niken Pujirahayu, Toshisada Suzuki, and Takeshi Katayama

Respone 5: yes it has been changed

Point 6: line 55, “α-amyrin” instead of “α-amyrine”,

Respone 6: yes it has been changed

Point 7: line 57, “dipterocarpol, a dammarane triterpene” instead of “dipterocarpol

dammarane triterpene” 

Respone 7: yes it has been fixed

Point 8: line 73, “The ether-soluble fraction” instead of “ether extract”,

Respone 8: ok it has been changed

Point 9: In Figure 1, the authors subtitled the structures with the letters (a) and (b),

but they did not mention them in the text. In the same figure, mangiferonic

acid should not be in italics

Respone 9: ok, it has been fixed

Point 10: In Table 1, the heading “Position” should be replaced by “Carbon”, 

Respone 10: yes it has been changed

Point 11: line 82, “Based on the NMR spectra data (Tables 1 and 2)” instead of

“Based on the NMR spectrum of the 1H and 13C tables”,

Respone 11: ok, it has been changed

Point 12: line 83, “cyclopropane moiety” instead of “C9-C19-C10 bond”,

Respone 12:yes it has been changed

Point 13: line 84, “ring can be seen in the 1H and 13C NMR spectra” instead of “bond

is seen in the 1H spectra and 13C spectra”

Respone 13: ok, it has been changed

Reviewer 2 Report

Please see the comments (five) in the PDF file

Author Response

Point1:   Line 44 …. and on the flora of the region

Respone: yes it has been fixed

Point 2: Line 46 …… and have non-functional sting

Respone: yes it has been fixed

Point 3: General comments for all the discussion session:The main factors that effect on the propolis composition is its botanical origin. As the bees do not visit only one plant for the preparation of propolis, the differences found between samples of propolis from different regions are mainly attributed to the differences in the flora and less to the species of the bees. So please make the appropriate corrections/wordings in all discussion.

Respone:  thank you, we will improve the discussion as you suggested

Point 4 : Line 172 , It is not 80 % of the propolis but 80 % of the total area in the chromatogram obtained at 254 nm.

Respone: yes it has been changed

Point 5: Please describe the extraction  steps of the resin M. indica that you had applied in order to take the HPLC-UV 254 nm) profile

Respone:  yes, sorry we forgot to explain the extraction method of M. indica 

resin, almost the same as the propolis extraction method, here is the method: (we will write it in the method)

M indica resin Extraction

The M. indica resin sample (2.0 g) was ground into fine powder then extracted three times with 20 mL 99% ethanol in a shaker at room temperature for 24 hours. After filtration using the same method as P1, ethanol extract from M. indica resin (EEM, 1.176 g) was obtained. The EEM was then applied to take 254 nm HPLC-UV profiles from M. indica resin and compared with EEP P1 and P2.

Thank you very much for your correction and suggestions for improving this manuscript

Regards,

Niken Pujirahayu, Toshisada Suzuki, Takeshi Katayama
